# Hemojuvelin deficiency promotes liver mitochondrial dysfunction and predisposes mice to hepatocellular carcinoma

Abdolamir Allameh[1,5,7], Nico Hüttmann [2,7], Edouard Charlebois[1,7], Angeliki Katsarou[1], Wen Gu[1], Konstantinos Gkouvatsos[1,6], Elisa Pasini[3], Mamatha Bhat[3], Zoran Minic[2], Maxim Berezovski [2], Maria Guido[4], Carine Fillebeen[1] & Kostas Pantopoulos [1✉]

Hemojuvelin (HJV) enhances signaling to the iron hormone hepcidin and its deficiency causes iron overload, a risk factor for hepatocellular carcinoma (HCC). We utilized $Hjv^{-/-}$ mice to dissect mechanisms for hepatocarcinogenesis. We show that suboptimal treatment with diethylnitrosamine (DEN) triggers HCC only in $Hjv^{-/-}$ but not wt mice. Liver proteomics data were obtained by mass spectrometry. Hierarchical clustering analysis revealed that Hjv deficiency and DEN elicit similar liver proteomic responses, including induction of mitochondrial proteins. Dietary iron overload of wt mice does not recapitulate the liver proteomic phenotype of $Hjv^{-/-}$ animals, which is only partially corrected by iron depletion. Consistent with these data, primary $Hjv^{-/-}$ hepatocytes exhibit mitochondrial hyperactivity, while aged $Hjv^{-/-}$ mice develop spontaneous HCC. Moreover, low expression of HJV or hepcidin (HAMP) mRNAs predicts poor prognosis in HCC patients. We conclude that Hjv has a hepatoprotective function and its deficiency in mice promotes mitochondrial dysfunction and hepatocarcinogenesis.

[1] Lady Davis Institute for Medical Research and Department of Medicine, McGill University, Montreal, QC, Canada. [2] Department of Chemistry and Biomolecular Sciences, University of Ottawa, Ottawa, ON, Canada. [3] Ajmera Transplant Program, and Division of Gastroenterology and Hepatology, University Health Network, Toronto, ON, Canada. [4] Department of Medicine, University of Padova, Padova, Italy. [5] Present address: Department of Clinical Biochemistry, Faculty of Medical Sciences, Tarbiat Modares University, Tehran, Iran. [6] Present address: Hôpital de Nyon, Nyon, Switzerland. [7] These authors contributed equally: Abdolamir Allameh, Nico Hüttmann, Edouard Charlebois. ✉email: kostas.pantopoulos@mcgill.ca

Hepatocellular carcinoma (HCC) represents the most predominant form of liver cancer, the second most frequent cause of cancer-related mortality worldwide[1]. The growing incidence of HCC is related to the high prevalence of liver diseases, such as chronic hepatitis C, alcoholic or non-alcoholic fatty liver disease[2]. These conditions are often associated with mild to moderate hepatic iron overload, which is thought to facilitate progression to end stage liver disease[3]. Moreover, HCC is a common complication in iron overload disorders such as hereditary hemochromatosis or transfusion-dependent thalassemia[4,5].

Systemic iron homeostasis is controlled by hepcidin, a peptide hormone that limits iron efflux to the bloodstream[6]. Hepcidin is produced in hepatocytes and operates by inactivating the iron exporter ferroportin in target cells, including tissue macrophages and duodenal enterocytes. The expression of hepcidin is induced by iron, inflammatory signals and other cues. Iron-dependent regulation of hepcidin involves bone morphogenetic proteins, mainly BMP6 and BMP2, which are secreted from liver sinusoidal endothelial cells and bind to BMP receptors on hepatocytes[7]. This activates the SMAD signaling cascade that promotes transcription of the hepcidin-encoding *HAMP* gene. Efficient BMP/SMAD signaling to hepcidin requires hemojuvelin (HJV), a BMP co-receptor[8].

Disruption of the *HJV* gene results in severe hepcidin deficiency, which allows hyperabsorption of dietary iron by duodenal enterocytes and unrestricted iron entry into plasma from erythrophagocytic tissue macrophages. These pathogenic responses underlie the development of juvenile hemochromatosis, a rare, early-onset form of hereditary hemochromatosis[9]. The most common form of hemochromatosis is linked to mutations in *HFE* that causes milder hepcidin deficiency and usually manifests after the fourth decade of life[10]. A hallmark of juvenile and adult variants of hemochromatosis is the gradual saturation of the iron-binding capacity of transferrin, the plasma iron carrier, and the accumulation of redox-active and toxic non-transferrin-bound iron (NTBI), which is readily taken up by hepatocytes and other tissue parenchymal cells.

Hjv$^{-/-}$ mice recapitulate the iron overload phenotype of juvenile hemochromatosis[11,12] but do not develop spontaneous liver injury at young age. Nevertheless, they exhibit oxidative stress in the liver, early activation of profibrogenic hepatic stellate cells and increased sensitivity to CCl$_4$-induced liver fibrosis[13]. Moreover, they develop liver fibrosis in response to chronic feeding a high iron diet[14].

Herein, we explored the sensitivity of Hjv$^{-/-}$ mice to diethylnitrosamine (DEN). We report that DEN promotes HCC in these animals even under suboptimal conditions, where wild type controls are not affected. Moreover, we demonstrate that Hjv deficiency triggers changes in the liver proteome that are reminiscent to those observed in DEN-treated wild type animals. Some but not all liver proteomic changes can be attributed to iron overload. We also show that low *HJV* expression correlates with reduced survival in HCC patients. Our data suggest that Hjv exhibits a hepatoprotective function which is partially dependent on its function as a regulator of iron metabolism.

## Results

### Sensitivity of Hjv$^{-/-}$ mice to DEN-induced hepatocarcinogenesis
We used Hjv$^{-/-}$ mice as a model of hemochromatosis to study the impact of iron overload on HCC pathogenesis following injection with DEN, an established hepatocarcinogen[15]. Groups of female Hjv$^{-/-}$ and isogenic wild type mice were treated with DEN or saline at the age of 2 weeks (Fig. 1a). The animals developed normally and were sacrificed after 24 weeks; no significant differences were found in body weights among genotypes and treatments at the beginning and endpoint of the experiment (Fig. S1). All DEN- and saline-treated Hjv$^{-/-}$ mice manifested significant ($p < 0.001$) and quantitatively indistinguishable increases in serum iron levels and transferrin saturation compared to wild type counterparts (Fig. S2a, b). TIBC values were not affected by genotype or treatment (Fig. S2c). As expected, Hjv mRNA expression was undetectable in Hjv$^{-/-}$ livers (Fig. 1b), while Hamp mRNA was significantly ($p < 0.001$) suppressed (Fig. 1c) and iron content significantly ($p < 0.001$) elevated (Fig. 1d). We did not observe any suppressive effects of DEN treatment on Hamp mRNA expression in female wild type mice (Fig. 1c), contrary to a previous report in which the sex of the mice was not specified[16].

Histopathological analysis showed that livers from saline-treated wild type ($n = 4$) or Hjv$^{-/-}$ ($n = 7$) mice had normal tissue architecture (Fig. 1e). The DEN treatment did not cause HCC in wild type mice in this experimental setting. There was only a small (229 μM) dysplastic focus with globules in a section of 1 out of 9 DEN-treated wild type mice. By contrast, 10 out of 11 DEN-treated Hjv$^{-/-}$ mice manifested several larger dysplastic foci and grade 1 or grade 2 HCC nodules; a representative section with a grade 2 HCC nodule is indicated with arrows in Fig. 1e. HCC development was also evident from abdominal ultrasound analysis of live animals at the age of 20 weeks (Fig. S3). These data suggest that Hjv$^{-/-}$ mice are sensitized to DEN-induced hepatocarcinogenesis.

### Liver proteomic profiles reveal predisposition of Hjv$^{-/-}$ mice to HCC
Representative liver samples of 5 mice from each of the 4 experimental groups (including those with visible HCC lesions) were subjected to LC-MS/MS analysis. A total of 2100 proteins were identified after removal of common contaminants. Only proteins having a label-free quantification (LFQ) intensity in at least 25% of all combined MS runs (20 samples measured in duplicates = 40) were considered for the analysis. LFQ intensities of duplicates were averaged and missing values were imputed on the level of technical replicates by drawing values from a normal distribution shifted to simulate low abundant proteins as previously described[17]. 1296 proteins remained for the subsequent analysis. The subcellular distribution of all 2100 identified and 1296 quantified liver proteins compared to the protein distribution in maximal detectable liver proteome[18] is shown in Fig. 2a.

The primary aim was to uncover differences in the liver proteome caused by Hjv deficiency and DEN treatment that account in combination for the development of HCC lesions. Principal component analysis (PCA) reveals that liver proteome profiles of control Hjv$^{-/-}$ mice cluster together with those of DEN-treated wild type and Hjv$^{-/-}$ mice (Fig. 2b). Significantly changed proteins were identified by a one-way ANOVA and $p$-values were adjusted by Benjamini-Hochberg-Correction (FDR 0.05). Individual differences between experimental groups were tested by Tukey's HSD test and a log2 fold-change threshold. The results show that the levels of 230 proteins are significantly changed between the groups (Fig. 2c).

To evaluate the influence of Hjv deficiency and DEN-treatment, Z-scored protein intensities of significantly altered proteins were subjected to hierarchical clustering (Fig. 2d), which revealed 8 main protein groups (clusters A-H). Combined quantitative protein data are provided in Supplementary Data 1. Each cluster shown in Fig. 2d was further analyzed by performing a Fisher's exact test with Gene Ontology and Reactome pathway annotations. Clusters C and D show an inverse response compared to clusters F-H, whereas up-(F-H) and down- (C and D) regulation occurs in all groups compared to wild type control mice.

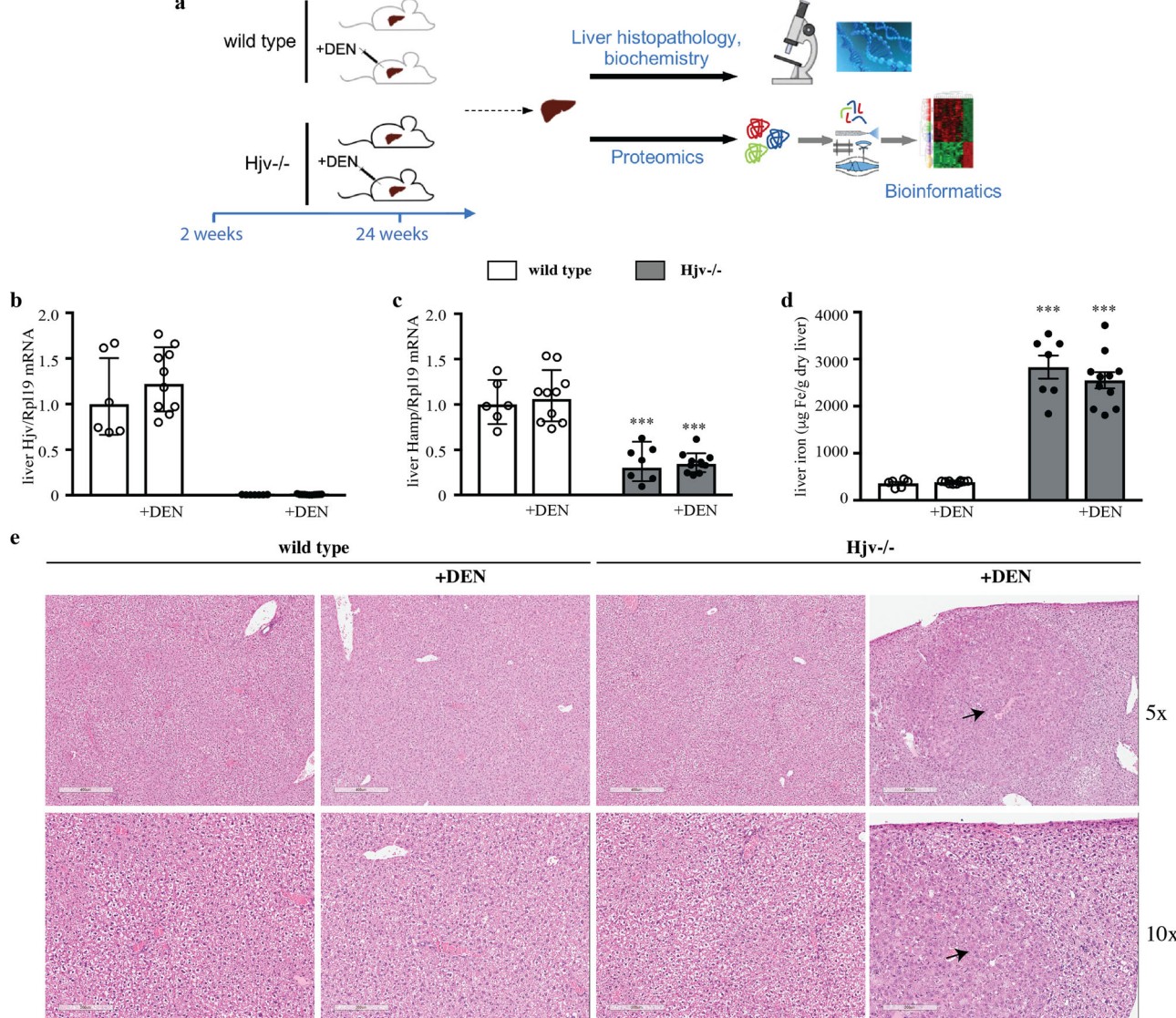

**Fig. 1 Hjv$^{-/-}$ mice develop early HCC in response to DEN treatment.** 2-week-old female wild type and Hjv$^{-/-}$ mice ($n = 8$–11 per group) were injected intraperitoneally with 50 mg/kg DEN or phosphate buffered saline. All animals were sacrificed after 24 weeks; livers were dissected and processed for biochemical, histological and proteomics analysis. **a** Schematic representation of the experimental design. **b** qPCR analysis of Hjv mRNA. **c** qPCR analysis of Hamp mRNA. **d** Quantification of liver iron. **e** Histopathological analysis of liver sections by H&E staining (magnifications: 5x and 10x). Arrows indicate HCC lesions. Data in **b**–**d** are presented as the mean ± SEM. Statistical analysis was performed by two-way ANOVA. Statistically significant differences across genotypes are indicated by ***($p < 0.001$).

Presumably, these proteins are regulated as a general stress response. Through functional annotation, clusters E-H are identified as mainly mitochondrial proteins, which is indicative of altered energy metabolism.

Clusters B and E are similarly inverse, such that both show a difference between Hjv$^{-/-}$ and wild type mice but not between DEN-treated and control group. These proteins are exclusively influenced by Hjv deficiency. Iron-dependent as well as metal-binding proteins can be found in both clusters B and E. These include L-ferritin (Ftl1), a positive control for an iron-inducible protein[19].

Clusters F and G show a steady upward trend from wild type control mice to DEN-treated Hjv$^{-/-}$ mice. In cluster E, the effects of DEN on both Hjv$^{-/-}$ and wild type subgroups do not appear particularly strong. However, the small changes triggered by DEN in Hjv$^{-/-}$ liver proteome are likely to accelerate formation of HCC lesions in these animals. Proteins within cluster D are annotated as of nuclear origin and are downregulated in livers of DEN-treated wild type but not Hjv$^{-/-}$ mice. Taken together, the data in Fig. 2 indicate that Hjv$^{-/-}$ mice are predisposed to hepatocarcinogenesis and exhibit a pro-HCC phenotype.

**Liver proteomic responses to Hjv deficiency and/or DEN treatment.** We first compared liver proteomic alterations between control wild type mice versus DEN-treated wild type and DEN-treated Hjv$^{-/-}$ mice by plotting the log2 fold-changes (Fig. 3a left). The graph exhibits a linear trend suggesting DEN treatment alone, and DEN treatment combined with Hjv deficiency exhibit similar responses in the proteomic profile. Commonly upregulated proteins include those involved in metabolic and mitochondrial processes (Fig. 3a right). The complete functional enrichment data of the DEN study are appended as Supplementary Data 2.

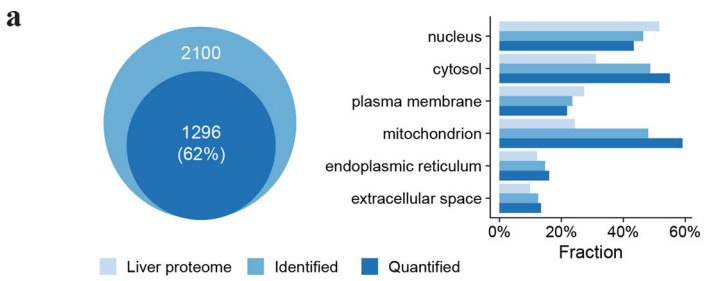

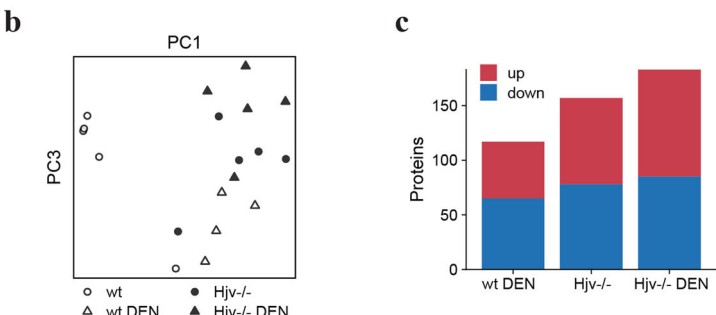

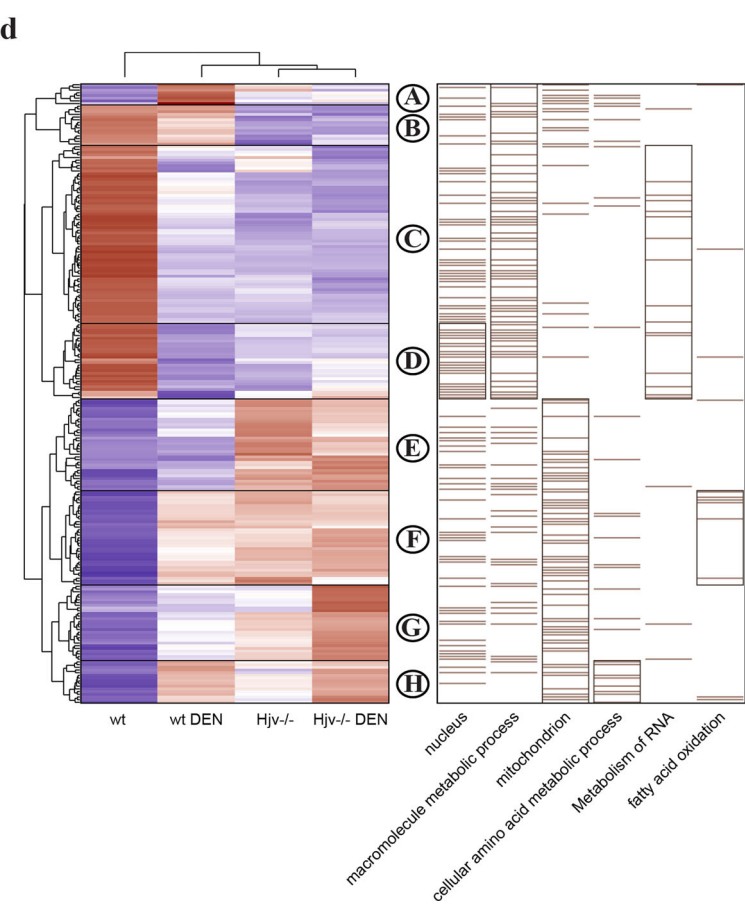

**Fig. 2 Alterations in the liver proteome of wild type and Hjv$^{-/-}$ mice following DEN treatment. a** Gene ontology analysis and subcellular distribution of quantified liver proteome compared to maximal detectable mouse liver proteome. **b** Principal component analysis (PCA) comparing the four mouse groups based on component 1 and 4, accounting for 26.6% and 7.6% of variability, respectively. **c** Number of differentially expressed proteins compared to wild type control mice. **d** Heat map of mean z-scored protein LFQ intensities of 230 differentially expressed proteins (ANOVA FDR < 0.05) after unsupervised hierarchical clustering. Enriched Gene Ontology terms among protein clusters are shown on the right. Bars represent position of annotated proteins; subclusters with highest enrichment (Fisher's exact test p-value) for each term are framed.

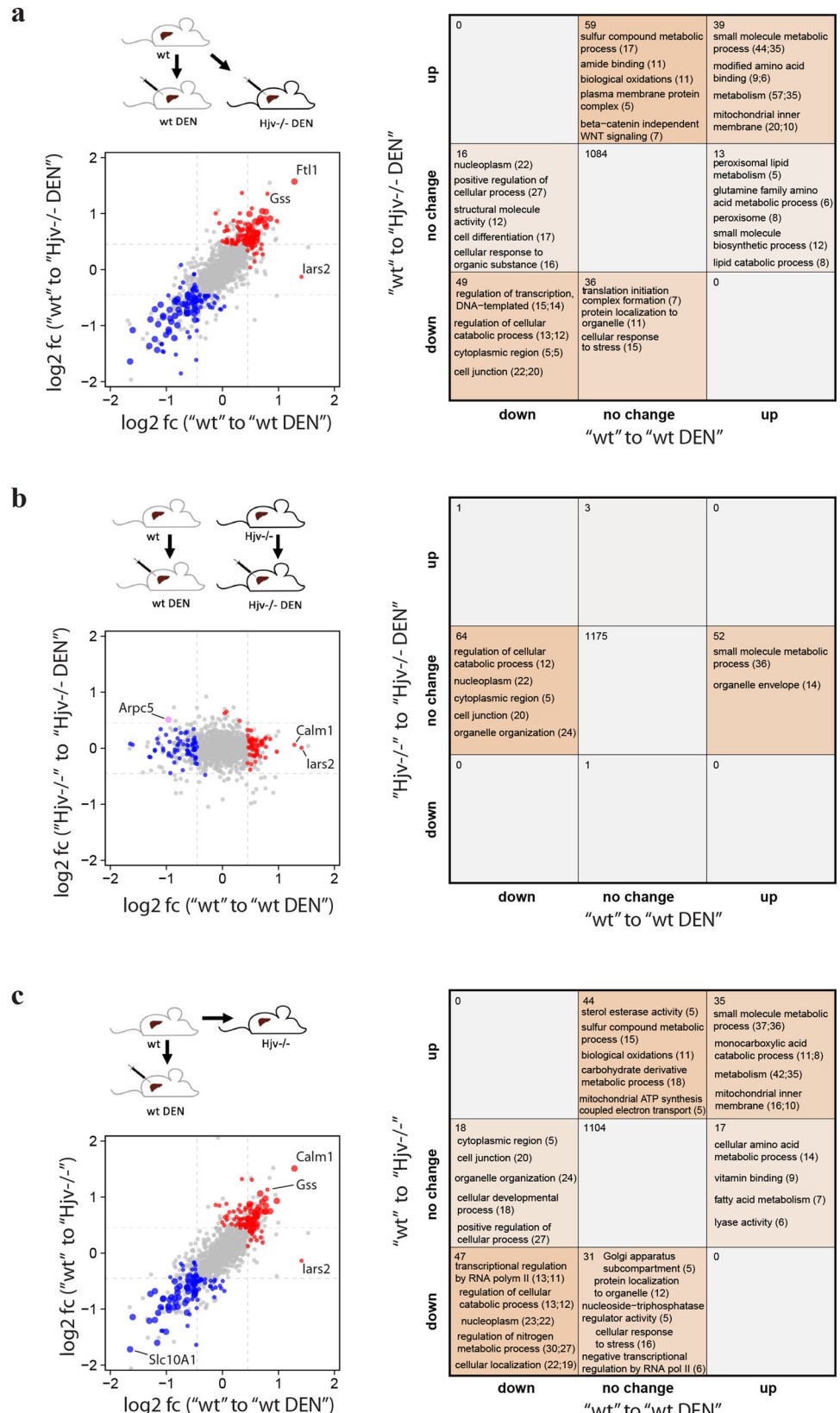

**Fig. 3 Comparative analysis of liver proteomes in wild type and Hjv$^{-/-}$ mice following DEN treatment. a** Liver proteomic changes between wild type versus DEN-treated wild type and Hjv$^{-/-}$ mice. **b** Liver proteomic changes in response to DEN treatment in wild type and Hjv$^{-/-}$ mice. **c** Liver proteomic changes in response to DEN treatment and Hjv deficiency. (Left) Log2 fold-changes (log2 fc) of differentially expressed proteins (Tukey $p < 0.05$, abs. log2 fold change >0.45). Red = upregulated, blue = downregulated, large point = significant in both. (Right) Number of significantly changed proteins and enriched annotations (Fisher's exact test $p < 0.05$) of respective protein sets.

Next, we compared wild type and Hjv$^{-/-}$ mice for liver proteomic alterations following DEN treatment (Fig. 3b). Absence of commonly deregulated proteins suggest that wild type and Hjv$^{-/-}$ mice respond differently to DEN. The small number of differentially regulated proteins in Hjv$^{-/-}$ mice shows that DEN elicits minor effects on the liver proteome of these animals, which is consistent with their pro-HCC phenotype.

Finally, we compared liver proteomic alterations between control wild type mice versus DEN-treated wild type and untreated control Hjv$^{-/-}$ mice (Fig. 3c). These results reveal a high similarity between responses to DEN treatment and Hjv deficiency, which further validates the pro-HCC phenotype of Hjv$^{-/-}$ mice. Commonly upregulated proteins are linked to metabolic and mitochondrial pathways. Commonly downregulated proteins are linked to cellular localization, nitrogen metabolism and RNA metabolism. In conclusion, the above data suggest that Hjv deficiency and the hepatocarcinogen DEN trigger similar changes in the liver proteome, which explains the predisposition of Hjv$^{-/-}$ mice to DEN-induced HCC.

**Aged Hjv$^{-/-}$ mice develop spontaneous HCC.** A group of wild type mice ($n = 5$; 3 males and 2 females) and a group of isogenic Hjv$^{-/-}$ mice ($n = 7$; 3 males and 4 females) were maintained on a standard rodent diet for 18 months after weaning. At the endpoint the mice were sacrificed, and livers were subjected to histopathological analysis. None of the wild type livers exhibited any HCC nodules. By contrast, we noted that 3 out of 3 male and 3 out of 4 female Hjv$^{-/-}$ mice had developed spontaneous HCC, which was not associated with liver fibrosis (Fig. 4). These findings suggest that Hjv deficiency eventually leads to HCC development without the need of a toxic stimulus.

**Is the predisposition of Hjv$^{-/-}$ mice to HCC due to iron overload?** Considering that hemochromatosis poses a risk factor for HCC, we sought to explore whether the pro-HCC phenotype of Hjv$^{-/-}$ mice is etiologically linked to hepatic iron overload. We addressed this by interrogating the liver proteome of 9-week-old male wild type and Hjv$^{-/-}$ mice, previously subjected to dietary iron manipulations for 5 weeks (Fig. 5a). We reasoned that if hepatic iron overload is the major HCC driver in Hjv$^{-/-}$ mice, their pro-HCC proteomic profile should be mitigated by dietary iron restriction. Moreover, dietary iron loading should recapitulate the pro-HCC proteomic profile in wild type mice. Iron loading of wild type mice was achieved by feeding a high-iron diet (HID). Conversely, Hjv$^{-/-}$ mice became relatively iron-depleted after feeding an iron-deficient diet (IDD). Control groups of wild type and Hjv$^{-/-}$ mice were fed a standard diet.

As expected, HID intake promoted a 3.5-fold induction in Hamp mRNA expression in wild type mice (Fig. 5b). Hamp mRNA levels were low in Hjv$^{-/-}$ mice on standard diet and were further suppressed below detection limit by IDD intake. Wild type mice on HID exhibited increased serum iron levels, transferrin saturation and serum ferritin, a marker of liver iron stores (Fig. S4). On the other hand, IDD intake failed to drop serum iron levels and transferrin saturation in Hjv$^{-/-}$ mice but caused a significant ($p < 0.01$) reduction in serum ferritin; these data are consistent with earlier observations[20].

The dietary iron manipulations were reflected in liver iron content (Figs. 5c, d). Thus, HID intake promoted a 11.6-fold increase ($p < 0.001$) in liver iron of wild type mice (Fig. 5c). Furthermore, Hjv$^{-/-}$ mice on IDD manifested a 3.6-fold reduction ($p < 0.001$) in liver iron compared to counterparts on standard diet. Interestingly, wild type mice on HID and Hjv$^{-/-}$ on IDD had a comparable liver iron content, which is also visible after histological staining of liver sections with Perls (Fig. 5d). It

should be noted that despite the comparable liver iron load, these animals exhibited dramatic differences in hepcidin expression since Hamp mRNA was maximized in wild type mice on HID and undetectable in Hjv$^{-/-}$ mice on IDD (Fig. 5b).

**Liver proteomic profiles of wild type and Hjv$^{-/-}$ mice following dietary iron manipulations.** Again, we analyzed representative liver samples of 5 mice from each of the 4 experimental groups by LC-MS/MS. The subcellular distribution of 1929 identified, and 1152 quantified proteins compared to the protein distribution in maximal detectable liver proteome[18] is shown in Fig. 6a. From 1152 quantified proteins, 199 were differentially expressed between groups (Fig. 6b). PCA shows that wild type mice on HID and Hjv$^{-/-}$ mice on standard diet have distinct liver proteomic profiles (Fig. 6c, d). Moreover, even though the liver proteome of Hjv$^{-/-}$ mice changes dramatically in response to relative iron depletion by IDD, it remains distinct compared to that of wild type mice (Fig. 6e). The hierarchical clustering analysis revealed 9 main protein groups (clusters a-h). Combined quantitative protein data can be found in Supplementary Data 1.

**Liver proteomic responses to Hjv deficiency and/or dietary iron manipulations.** We analyzed liver proteomic alterations between control wild type mice (on standard diet) versus dietary iron-loaded wild type mice (on HID) and genetically iron-loaded Hjv$^{-/-}$ mice (on standard diet); the data are shown in Fig. 7a. Most proteins respond differently between dietary iron loading and Hjv deficiency. This suggests that a substantial number of liver proteomic responses to Hjv deficiency are unrelated to iron overload. It is also possible that the liver proteomic disparities reflect differences in iron distribution between dietary and genetic iron loading. Interestingly, mitochondrial proteins are upregulated in both settings. The complete functional enrichment data of the dietary iron study can be found in Supplementary Data 3.

We next compared liver proteomic alterations among control wild type mice (on standard diet) versus genetically iron-loaded Hjv$^{-/-}$ mice (on standard diet) and dietary iron-depleted Hjv$^{-/-}$ mice (on IDD). The data in Fig. 7b shows large number of proteins being reversed after dietary iron depletion in Hjv$^{-/-}$ mice. This suggests that dietary iron depletion can partially but not fully correct the liver proteome of Hjv$^{-/-}$ mice.

Finally, on Fig. 7c we compared liver proteomic alterations between control wild type mice (on standard diet) versus iron-loaded wild type (on HID) and iron-depleted Hjv$^{-/-}$ mice (on IDD), which exhibit comparable liver iron content. Only 7 and 11 proteins are commonly up- and down-regulated between both conditions, respectively. These data further suggest that many liver proteomic responses to Hjv deficiency are unrelated to iron.

**Liver proteomic responses to DEN and iron.** We compared the datasets of the experiments described in Figs. 1–3 and 5–7 to identify which proteins were deregulated in response to both DEN and iron. In total, 998 of proteins were quantified in both studies (Fig. 8a). We then compared liver proteomic differences among wild type and Hjv$^{-/-}$ mice between the two datasets (Fig. 8b). Several proteins were differentially expressed in both studies, but only 9 and 4 proteins were consistently and significantly up- or down-regulated, respectively. Presumably, this is related to the age and sex differences of the mice used in the two studies (26-week-old female versus 9-week-old male), but also to our stringent cutoffs. Pathway analysis (right panel) reveals that the 9 upregulated proteins in Hjv$^{-/-}$ livers are involved in mitochondrial processes, metabolism, glutathione conjugation/metabolism, fatty acid catabolism and lipase activity, while the 4 downregulated proteins are involved in stress responses (Fig. 8b).

**a**

| Genotype | Sex | Number of mice | Number of mice with HCC |
|---|---|---|---|
| wild type | male | 3 | 0 |
| wild type | female | 2 | 0 |
| Hjv-/- | male | 3 | 3 |
| Hjv-/- | female | 4 | 3 |

**b**

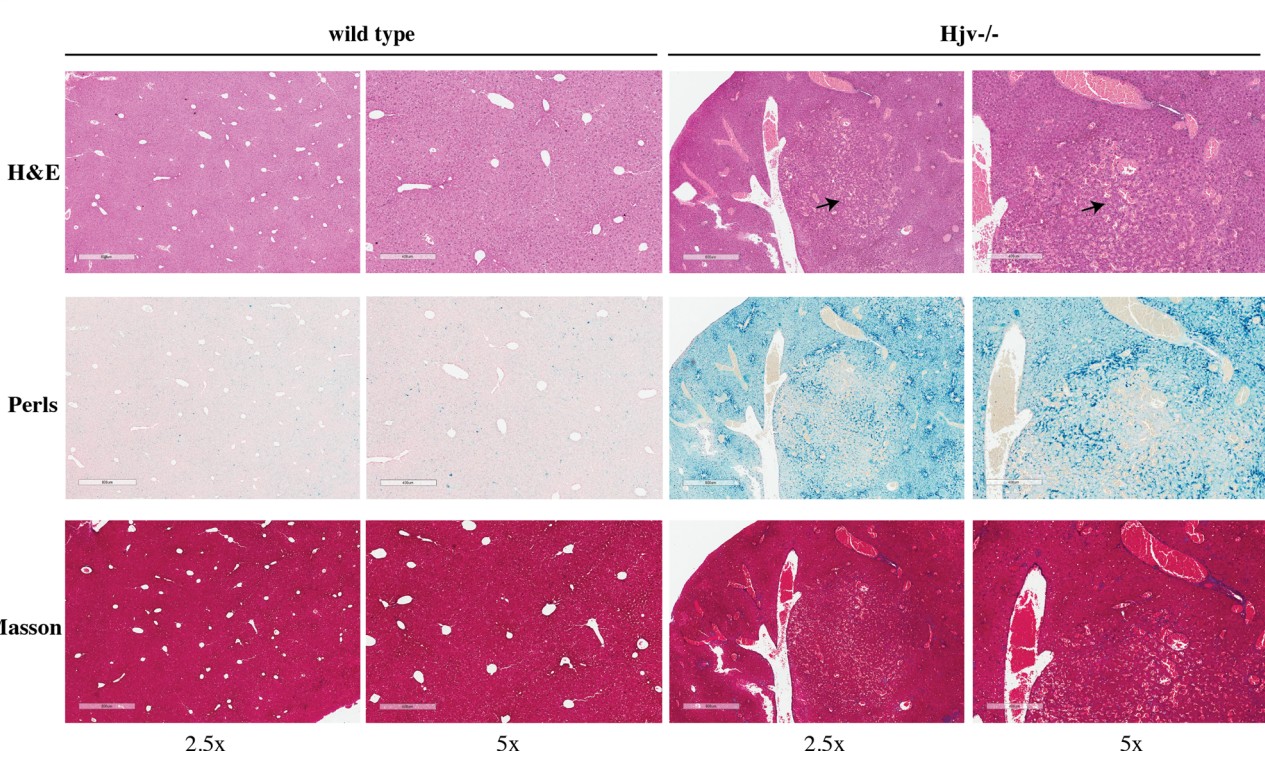

**Fig. 4 Histopathological analysis of livers from aged wild type and Hjv$^{-/-}$ mice. a** Genotypes, sex, numbers of mice and numbers of mice with histologically documented HCC. **b** Representative liver sections stained with H&E, Perls or Masson's trichrome (magnifications: 2.5× and 5×). HCC lesions are indicated by arrows.

We then analyzed the responses of all 13 differentially regulated proteins to DEN or dietary iron manipulations in wild type and Hjv$^{-/-}$ mice. The heatmap in Fig. 8c shows that Ftl1, Gstm1, Glyat, Sdha, Ces1d, Idh2, Nipsnap1 and Ces1g are all induced by DEN, Hjv deficiency or iron; moreover, they tend to be "corrected" by iron depletion in Hjv$^{-/-}$ mice (except Ces1d). Expression of Adss is only induced by Hjv deficiency and remains unaffected by iron or DEN. Finally, Hnrnpdl, Ubxn1, Lap3 and Fabp1 are suppressed by Hjv deficiency or DEN and do not appear to significantly respond to dietary iron.

**Hjv deficiency promotes mitochondrial hyperactivity in hepatocytes.** Our proteomics data identified a strong induction of mitochondrial proteins in Hjv deficient livers, which is indicative of altered energy metabolism. To validate these findings and explore functional implications, we analyzed mitochondrial respiration in cultured primary hepatocytes from wild type and Hjv$^{-/-}$ mice using the Seahorse assay. We further investigated the role of iron by treating wild type and Hjv$^{-/-}$ cells with ferric ammonium citrate (FAC) or the iron chelator desferrioxamine (DFO), respectively. Hjv$^{-/-}$ hepatocytes exhibited dramatically increased oxygen consumption rate (OCR) for basal respiration, spare respiratory capacity, maximal respiratory capacity, proton leak and ATP production (Fig. 9). These effects were blunted

following DFO treatment, indicating strong dependence on iron. However, acute iron treatment of wild type hepatocytes failed to mimic the respiratory phenotype of Hjv$^{-/-}$ hepatocytes.

**HJV and HAMP expression in human HCC.** Our findings in Hjv$^{-/-}$ mice raise the intriguing hypothesis that the expression of *HJV* and possibly also its downstream target *HAMP* may have a prognostic value in human HCC. To address this, we used KMplotter, a web-based tool that enables survival analysis across multiple cancers and datasets[21]. Patient data from the TCGA HCC dataset were split into two groups according to auto selection of the best cut-off for *HJV* and *HAMP* mRNAs (RNAseq probe IDs 148738 and 57817, respectively). To this end, we computed all possible cut off values between the lower and upper quartiles and selected the best performing threshold as cut off. In addition to the p-value, the False Discovery Rate (FDR) was computed to correct for multiple hypothesis testing[22]. We first ran multivariate overall survival analysis based on the high versus low expression of *HJV* mRNA in tumors. The two groups were compared by a Kaplan-Meier survival plot (Fig. 10a), and the hazard ratio with 95% confidence intervals and log-rank p-value were calculated. A similar comparison was also performed for expression of *HAMP* mRNA in tumors (Fig. 10b).

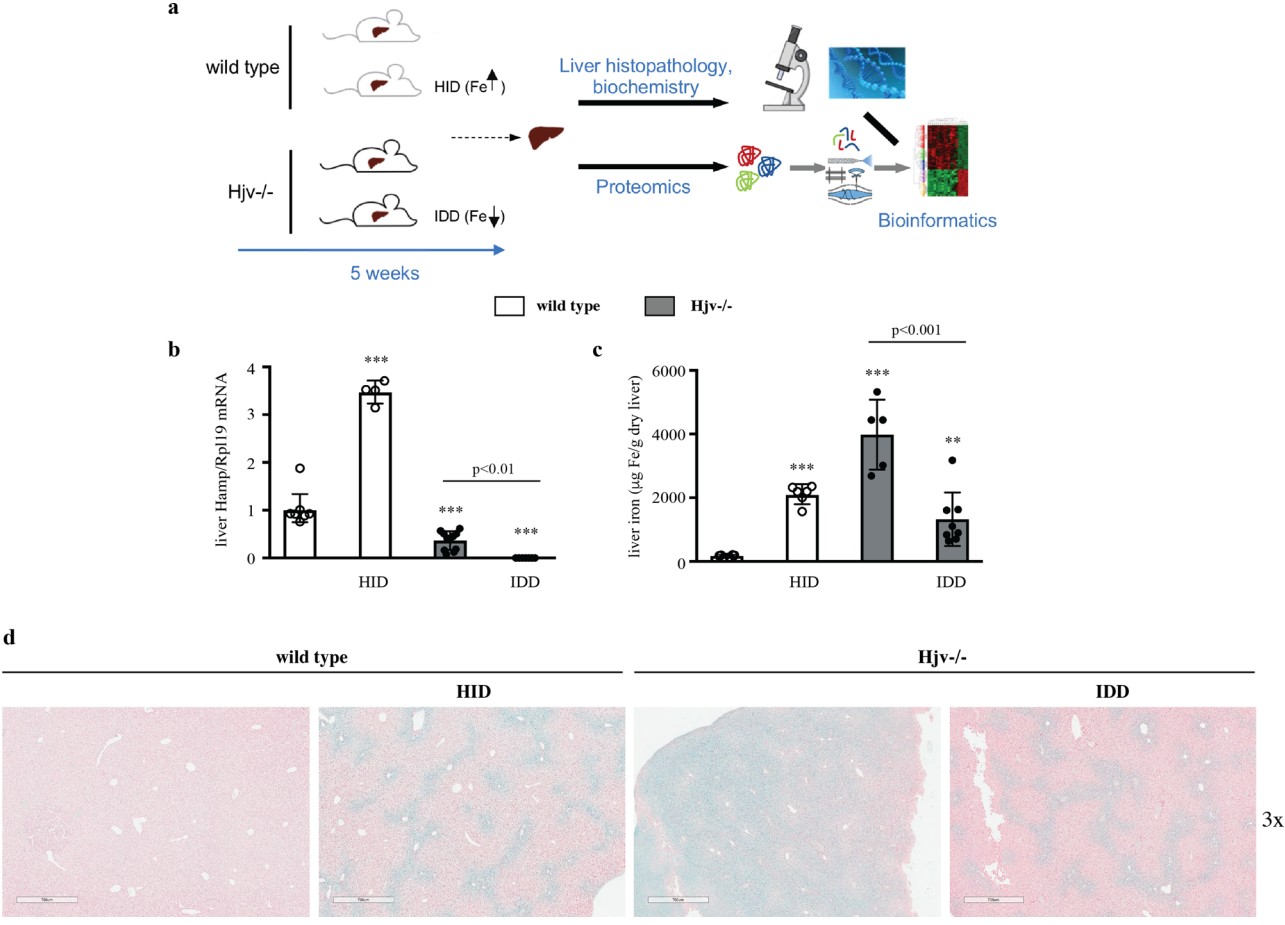

**Fig. 5 Dietary iron manipulations in wild type and Hjv$^{-/-}$ mice.** 4-week-old male wild type and Hjv$^{-/-}$ mice ($n = 8$–10 per group) were fed a standard diet, a high iron diet (HID) or an iron-deficient diet (IDD). All animals were sacrificed after 5 weeks; livers were dissected and processed for biochemical, histological and proteomics analysis. **a** Schematic representation of the experimental design. **b** qPCR analysis of Hamp mRNA. **c** Quantification of liver iron. **d** Perls Prussian blue staining in liver sections (magnification: 3x). Data in (**b**–**c**) are presented as the mean ± SEM. Statistical analysis was performed by two-way ANOVA. Statistically significant differences across genotypes are indicated by **($p < 0.01$) or ***($p < 0.001$).

Overall survival analysis was evaluated using available RNAseq data for *HJV* and *HAMP* from 364 patients. Presence of *HJV* in HCC was associated with a significantly decreased hazard ratio of 0.57 (95% confidence interval 0.39–0.82, log-rank $p = 0.0023$), as shown in Fig. 10a. Likewise, expression of *HAMP* in HCC correlated with a significantly decreased hazard ratio of 0.62 (95% confidence interval 0.41–0.92, log-rank $p = 0.016$), as shown in Fig. 10b. These data suggest that expression of *HJV* and *HAMP* are significantly prognostic of overall survival in HCC patients.

We finally explored whether low *HJV* or *HAMP* expression in HCC may represent less differentiated cancers. The oncofetal marker genes *E2F1*, *FOXM1*, *TOP2A*, *ECT2*, *HELLS* and *UHRF1*[23] in the TCGA HCC dataset were likewise analyzed using KMplotter. High expression of all these these genes was associated with poor survival (Fig. S5). Interestingly, there was a weak but statistically significant inverse correlation between *HJV* or *HAMP* mRNAs and all oncofetal markers (Fig. 10c); Spearman's rank correlation coefficients and p values are provided in Table S1. These findings indicate that decreased *HJV* and *HAMP* expression in aggressive tumors is only partially a consequence of low tumor cell differentiation.

## Discussion

Iron overload is a known independent risk factor for HCC incidence and progression[4,24]. Thus, HCC is frequently observed in adults with HFE-related hereditary hemochromatosis[25,26] but also

sporadically in younger juvenile hemochromatosis patients with *HJV* mutations[27], who typically present with cardiomyopathy.

To explore underlying mechanisms, female Hjv$^{-/-}$ mice, a model of juvenile hemochromatosis, and isogenic (in C57BL/6 background) wild type control animals were treated with the hepatocarcinogen DEN. HCC exhibits sexual dimorphism with substantially higher prevalence in male patients[28] and mouse models[29]. A single injection of DEN typically promotes HCC within 42 weeks in male wild type C57BL/6 mice[30]. Consistent with this finding, we did not observe any nodules in our DEN-treated female wild type C57BL/6 animals after 24 weeks. However, the vast majority (10 out of 11) of the female Hjv$^{-/-}$ mice manifested HCC within 24 weeks of DEN treatment, demonstrating increased sensitivity of this mouse model to hepatocarcinogenesis. Similar data were recently reported with hepatocyte-specific Fbxl5$^{-/-}$ mice, a model of hepatocellular iron overload due to overexpression of the cellular iron sensor IRP2[31].

Importantly, we also noted that aged Hjv$^{-/-}$ mice tend to develop spontaneous HCC (Fig. 4). This finding is reminiscent of spontaneous hepatocarcinogenicity in response to long-term dietary iron overload that has been reported in rats[32,33] and transgenic mice engineered to express the hepatitis C virus (HCV) polyprotein[34]. The latter develop HCC anyway at the age of 16 months[35], and it appears that iron overload accelerates this process.

HCC pathogenesis is associated with extensive genetic reprogramming in liver cells, which is characterized by heterogeneity

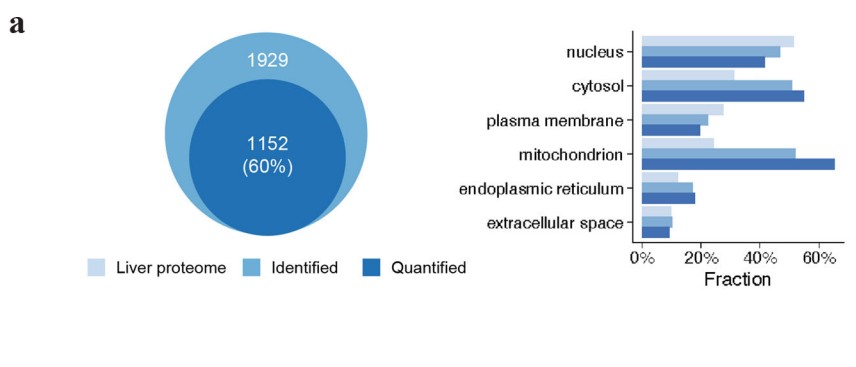

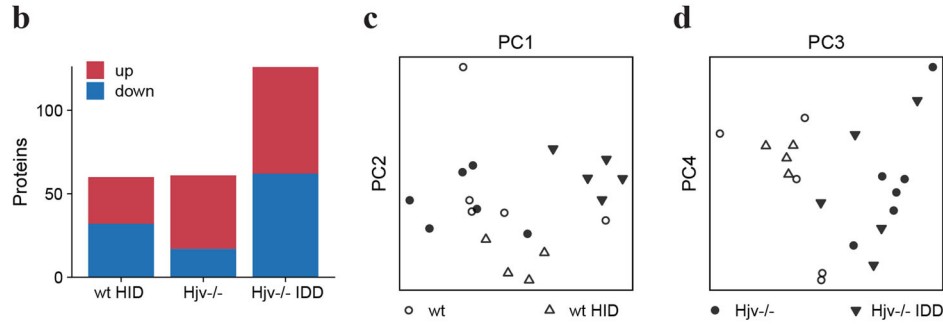

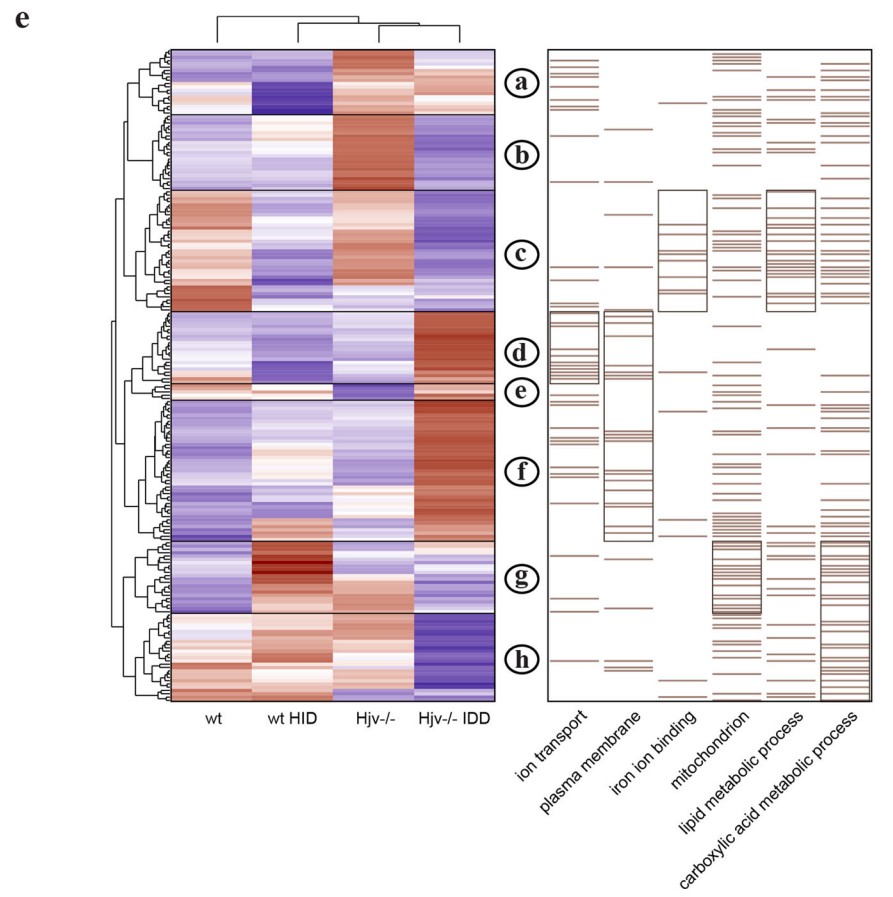

and distinct molecular features[36–39]. To better understand the impact of iron overload, we analyzed and compared liver proteomic profiles of control and DEN-treated Hjv[−/−] and wild type mice. The coverage of the mouse liver proteome in our whole-tissue analysis was ~2,000 proteins, mostly soluble from cytosol,

nucleus and mitochondria (Figs. 2a and 6a). Membrane proteins were underrepresented due to extraction conditions. Substantially higher coverage of the mouse liver proteome (~11,500 proteins) is only possible when the analysis is performed in purified liver cell populations following liver perfusion[18]. This approach has

**Fig. 6 Liver proteomic profiles of wild type and Hjv$^{-/-}$ mice after dietary iron manipulations. a** Gene ontology analysis and subcellular distribution of quantified liver proteome compared to maximal detectable mouse liver proteome. **b** Number of differently expressed proteins compared to wild type mice on standard diet. **c** Principal component analysis (PCA) of the four mouse groups based on component 1 and 2 (top), accounting for 25.5% and 11.3% of variability, and **d** components 3 and 4 (bottom), accounting for 9% and 6.8% of variability, respectively. **e** Heat map of mean z-scored protein LFQ intensities of 199 differentially expressed proteins (ANOVA FDR < 0.05) after unsupervised hierarchical clustering. The order of sample groups has been manually adjusted. Enriched Gene Ontology terms among protein clusters are shown on the right. Bars represent protein annotations and enriched clusters of proteins are framed.

yielded the hitherto maximal detectable mouse liver proteome. A detailed distribution of liver proteins identified herein and in[18] is illustrated in Supplementary Data 4.

Surprisingly, we found that the liver proteome of untreated control Hjv$^{-/-}$ mice clusters together with that of DEN-treated Hjv$^{-/-}$ and wild type mice (Fig. 2b). Thus, the expression of many proteins is controlled similarly by Hjv deficiency or DEN treatment (Fig. 3c). Common upregulated proteins are mainly mitochondrial, which are known to be involved in energy metabolism and stress response pathways. In fact, hepatocytes from Hjv$^{-/-}$ mice exhibit increased respiratory capacity and energy production (Fig. 9), which is consistent with recent data obtained with livers from Hfe$^{-/-}$ mice, another model of hemochromatosis[40]. Notably, iron-induced HCC development in the transgenic HCV mouse model was likewise associated with mitochondrial dysfunction[34].

Although the predisposition of Hjv$^{-/-}$ mice to HCC can be largely attributed to hepatic iron overload, potential iron-independent effects of Hjv deficiency cannot be excluded. To distinguish between these two non-mutually exclusive scenarios, we compared liver proteomic profiles of wild type and Hjv$^{-/-}$ mice following dietary iron manipulations. We specifically explored whether amelioration of hepatic iron overload by dietary iron restriction can correct the pro-HCC liver proteomic phenotype in Hjv$^{-/-}$ mice. We further examined whether dietary iron overload triggers pro-HCC changes in the liver proteome of wild type mice. However, we found that iron-loaded wild type mice and relatively iron-depleted Hjv$^{-/-}$ counterparts with a comparable liver iron content (Fig. 5), exhibit remarkable differences in their liver proteomic profiles (Figs. 6, 7). These data suggest that iron overload may not fully account for the proteomic changes manifested in the liver of Hjv$^{-/-}$ mice. While the induction of mitochondrial protein expression could be mimicked by dietary iron overload in wild type mice, in vitro iron loading of cultured wild type primary hepatocytes failed to recapitulate the increased mitochondrial oxygen consumption rate documented in primary Hjv$^{-/-}$ hepatocytes (Fig. 9). Nevertheless, iron chelation corrected mitochondrial hyperactivity in Hjv$^{-/-}$ hepatocytes, which may develop as an adaptation to chronic rather than acute iron overload.

Iron appears to be a key inducer of glutathione metabolism proteins, which are also upregulated following exposure to DEN (Fig. 8b, c). Conceivably, these responses are protective against ferroptosis, an iron-dependent form of programmed cell death that involves lipid peroxidation and disturbances in glutathione metabolism[41].

Some of the variable liver proteomic responses to dietary and genetic iron loading in wild type and Hjv$^{-/-}$ mice, respectively, may be related to the different iron distribution. It is well established that in hereditary hemochromatosis the insufficiency of hepcidin allows high expression of the iron exporter ferroportin in Kupffer cells. Thus, while hepatocytes accumulate excess iron, Kupffer cells fail to retain iron and become paradoxically iron-deficient[6,10]. By contrast, mice fed high-iron diets develop iron overload in both hepatocytes and Kupffer cells[42].

On the other hand, it has been reported that dietary iron-loaded wild type mice and genetically iron-loaded Hjv$^{-/-}$ mice exhibit similar upregulation of ferroptosis markers[43]. Nevertheless, our data

suggest that the pro-HCC phenotype of Hjv$^{-/-}$ mice is dominant and not mitigated by the apparent sensitivity of these animals to ferroptosis. While our findings are consistent with an important role of hepatic iron overload in the predisposition of Hjv$^{-/-}$ mice to HCC, they also highlight potential iron-independent pathogenic contributions of Hjv deficiency. These remain to be characterized at the molecular level. It should, however, be noted that our Hjv$^{-/-}$ mice on iron-deficient diet were not fully iron-depleted but retained significant residual iron overload (Fig. 5). Effective iron depletion in these animals can only be achieved when low dietary iron intake is combined with phlebotomies[44]. To better dissect pathogenic mechanisms triggered by iron overload vs Hjv deficiency, it will be important to utilize the model of DEN-induced hepatocarcinogenesis with dietary iron-loaded wild type mice and fully iron-depleted Hjv$^{-/-}$ mice.

The data in Fig. 8c identify a group of proteins that are deregulated in response to DEN, Hjv deficiency or iron, and likely contribute to hepatocarcinogenesis. Some of them (Ftl1, Gstm1, Glyat, Sdha, Ces1d, Idh2 and Nipsnap1) can be corrected by relative iron depletion of Hjv$^{-/-}$ mice, while stress response proteins (Hnrnpdl, Ubxn1, Lap3 and Fabp1) appear to be suppressed by Hjv deficiency or DEN in an iron-independent manner. Pathway analysis demonstrates involvement of the above proteins in mitochondrial pathways, metabolism, glutathione conjugation/metabolism, fatty acid catabolism, lipase activity and stress responses. This pattern is indicative of metabolic reprogramming and shows remarkable similarities to a "metabolic" subtype of human HCC related to hepatitis B virus (HBV) that was identified by proteomic studies[38]. Our findings reinforce the concept that mitochondrial hyperactivity and enhanced energy metabolism are possible drivers of hepatocarcinogenesis[45,46].

The clinical relevance of our studies with the Hjv$^{-/-}$ mouse model is illustrated by the identification of a prognostic value of *HJV* mRNA expression in human HCC (Fig. 10a). Thus, decreased *HJV* mRNA levels in tumors is associated with poor prognosis. Similar results were obtained with *HAMP* mRNA (Fig. 10b) and corroborate recent findings[47]. Reduced *HJV* expression has been previously reported in hepatoma cell lines and HCC tissues[48], while the suppression of *HAMP* in HCC is well established[16,47,49–51]. The weak but significant inverse correlation of *HJV* and *HAMP* with expression of oncofetal markers (Fig. 10c), which are predictors of low survival (Fig. S5), indicates that at least part of *HJV* and *HAMP* suppression in HCC is likely a consequence of poor cell differentiation giving rise to more aggressive tumors. Nevertheless, a pathogenic contribution of HJV deficiency in HCC cannot be excluded by the analysis of the clinical dataset and, furthermore, is strongly supported by the experiments in the mouse models. Overall, our data raise the possibility that targeting iron metabolism via the HJV/HAMP axis could potentially improve outcomes in HCC.

## Methods

**Animal housing.** Hjv$^{-/-}$ mice in C57BL/6 background[20] and isogenic wild type controls were housed in macrolone cages (up to 5 mice/cage, 12:12 h light-dark cycle: 7 am–7 pm; 22 ± 1 °C, 60 ± 5% humidity) and were allowed *ad libitum* access to chow and drinking water. The mice were fed a standard diet (containing

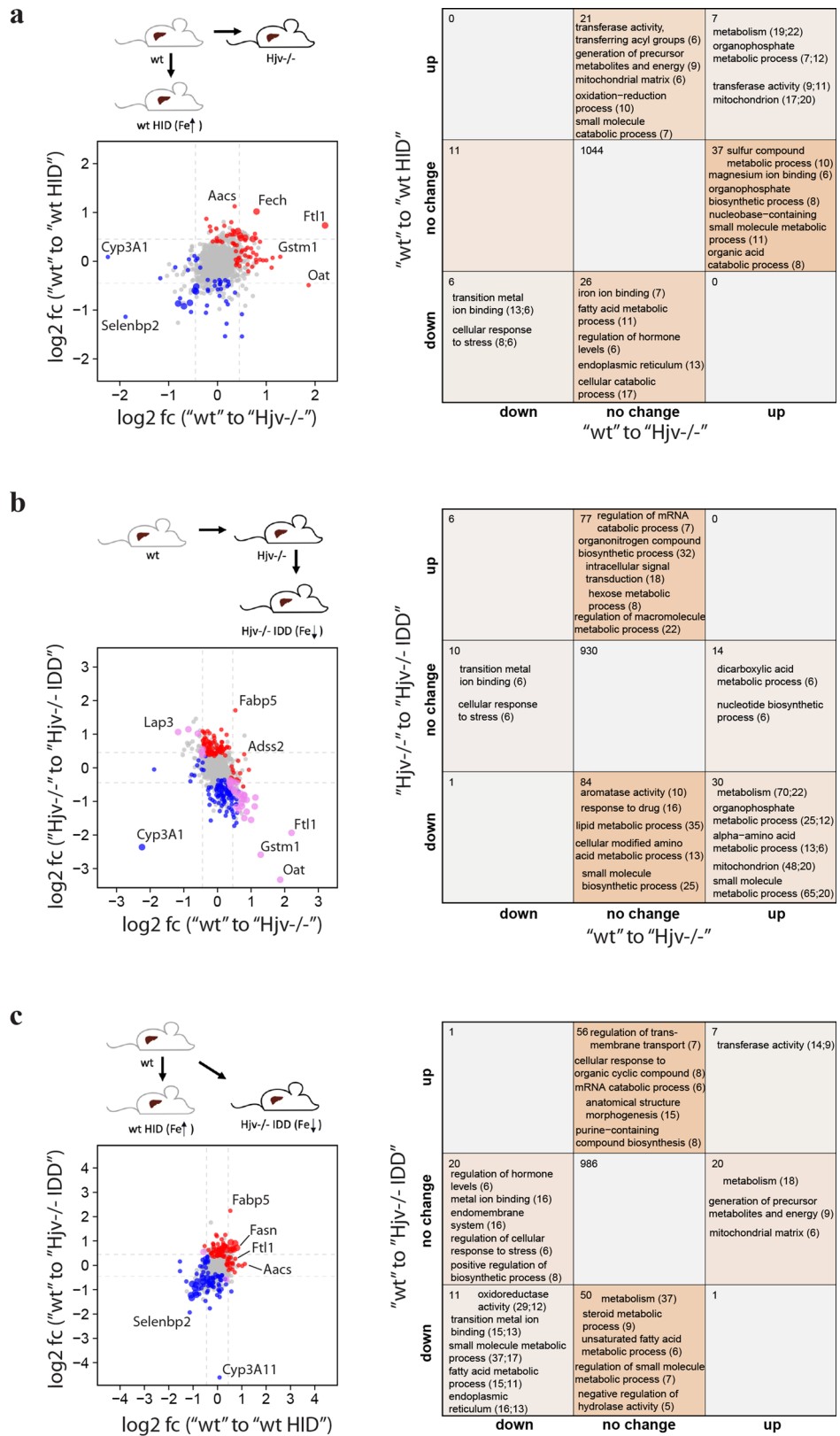

**Fig. 7 Comparative analysis of liver proteomes in wild type and Hjv$^{-/-}$ mice following dietary iron manipulations. a** Liver proteomic changes in response to high iron diet (HID) versus Hjv deficiency. **b** Liver proteomic changes in response to Hjv deficiency and relative dietary iron depletion. **c** Liver proteomic changes between wild type mice on standard diet versus wild type mice on HID and Hjv$^{-/-}$ mice on IDD. (Left) Log2 fold-changes (log2 fc) of differentially expressed proteins (Tukey $p < 0.05$, abs. log2 fold change >0.35). Red = upregulated, blue = downregulated, large point = significant in both. (Right) Number of significantly changed proteins and enriched annotations (Fisher's exact test $p < 0.05$) of respective protein sets.

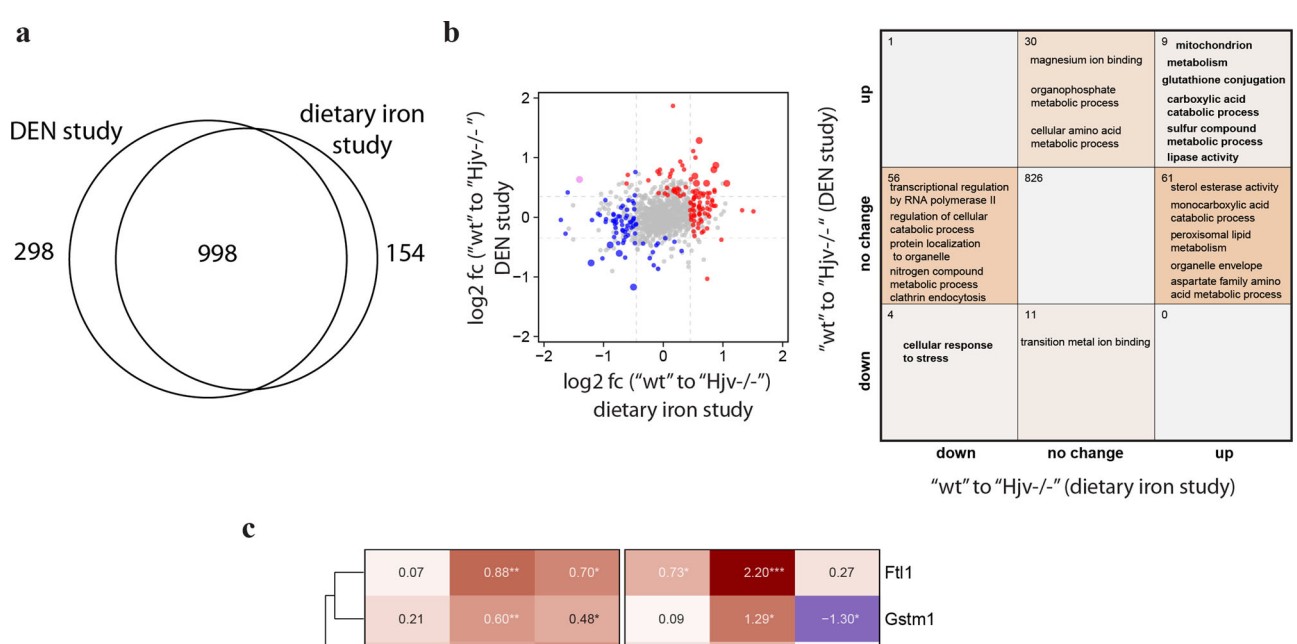

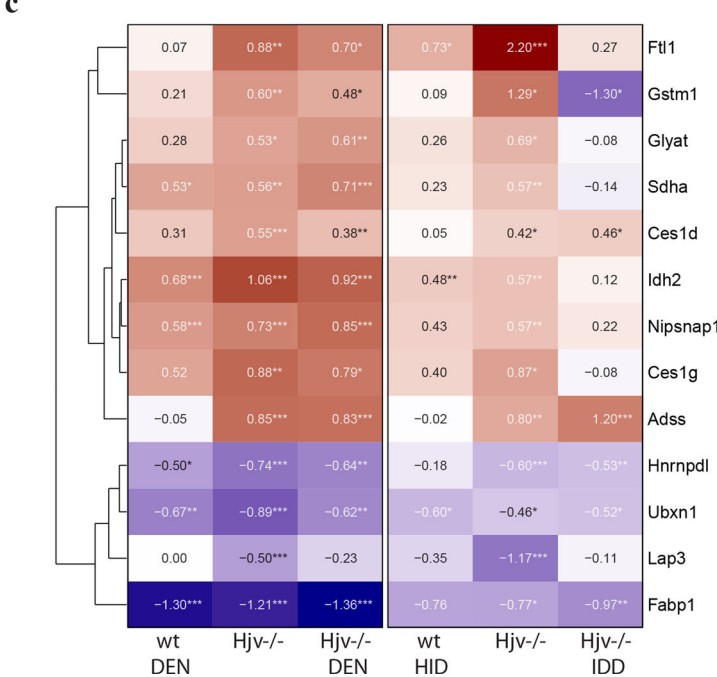

**Fig. 8 Mitochondrial proteins and proteins involved in metabolism and glutathione pathways are induced by Hjv deficiency, DEN or iron, and corrected by iron depletion. a** Venn diagram comparing overlapping quantified proteins in the DEN and dietary iron studies. **b** Liver proteomic changes among wild type and Hjv$^{-/-}$ mice between the DEN and dietary iron studies. (Left) Log2 fold-changes (log2 fc) of differentially expressed proteins (Tukey $p < 0.05$, abs. log2 fold change >0.35). Red = upregulated, blue = downregulated, large point = significant in both. (Right) Number of significantly changed proteins and enriched annotations (Fisher's exact test $p < 0.05$) of respective protein sets. **c** Heat map of log2 fold-changes compared to wild type control mice of 13 commonly deregulated proteins.

200 ppm iron); when indicated, the mice were fed similar diets only differing in iron content (iron-deficient diet: 2–6 ppm iron; high-iron diet: 2% carbonyl iron)[52]. At the endpoints, the animals were sacrificed by $CO_2$ inhalation. Blood was collected via cardiac puncture and serum was prepared, snap-frozen in liquid nitrogen and stored at −80 °C. Livers were harvested and a small portion of the tissue was fixed in 10% buffered formalin and embedded in paraffin for histopathological analysis. The remaining tissue was cut into pieces, snap-frozen in liquid nitrogen and stored at −80 °C for biochemical analysis. Experimental procedures were approved by the Animal Care Committee of McGill University (protocol 4966).

**DEN treatments**. 2-week-old female animals were injected once intraperitoneally with 50 mg/kg DEN dissolved in sterile phosphate buffered saline. A total volume of 0.6 ml DEN was injected to both sides of the peritoneal area to avoid any spillover. Mice in control groups were injected with equal volume of phosphate buffered saline. All injections were done in a quarantine room under aseptic conditions. After 72 h the animals were ear-tagged, divided into cages, and transferred to the main animal facility, where they remained for 24 weeks. Body weight was monitored at the beginning of the treatment and at the endpoint. Abdominal sonography was performed after 12 and 20 weeks, using in vivo

imaging system (VEVO 770 high frequency ultrasound). The experiment was terminated after 24 weeks and the mice were euthanized, as described above.

**Liver histopathology**. Liver specimens were stained with H&E to analyze tissue architecture, and with Perls to visualize iron deposits.

**Serum iron biochemistry**. Serum iron, total iron binding capacity (TIBC) and ferritin were measured on a Roche Hitachi 917 Chemistry Analyzer. Transferrin saturation was calculated from the ratio of serum iron and TIBC.

*Liver iron quantification*. Non-heme liver iron was measured by the ferrozine assay[53].

**Quantitative real-time PCR (qPCR)**. Liver RNA was analyzed by qPCR as previously described[54,55] by using gene-specific primers for: Hamp1 (forward AAG-CAGGGCAGACATTGCGAT; reverse CAGGATGTGGCTCTAGGCTATGT), Hjv (forward ATCCCCATGTGCGCAGTTTT; reverse GCTGGTGGCCTGGA-CAAA) and ribosomal protein L19 (Rpl19) (forward AGGCATATGGGCA-TAGGGAAGAG; reverse TTGACCTTCAGGTACAGGCTGTG). Relative mRNA

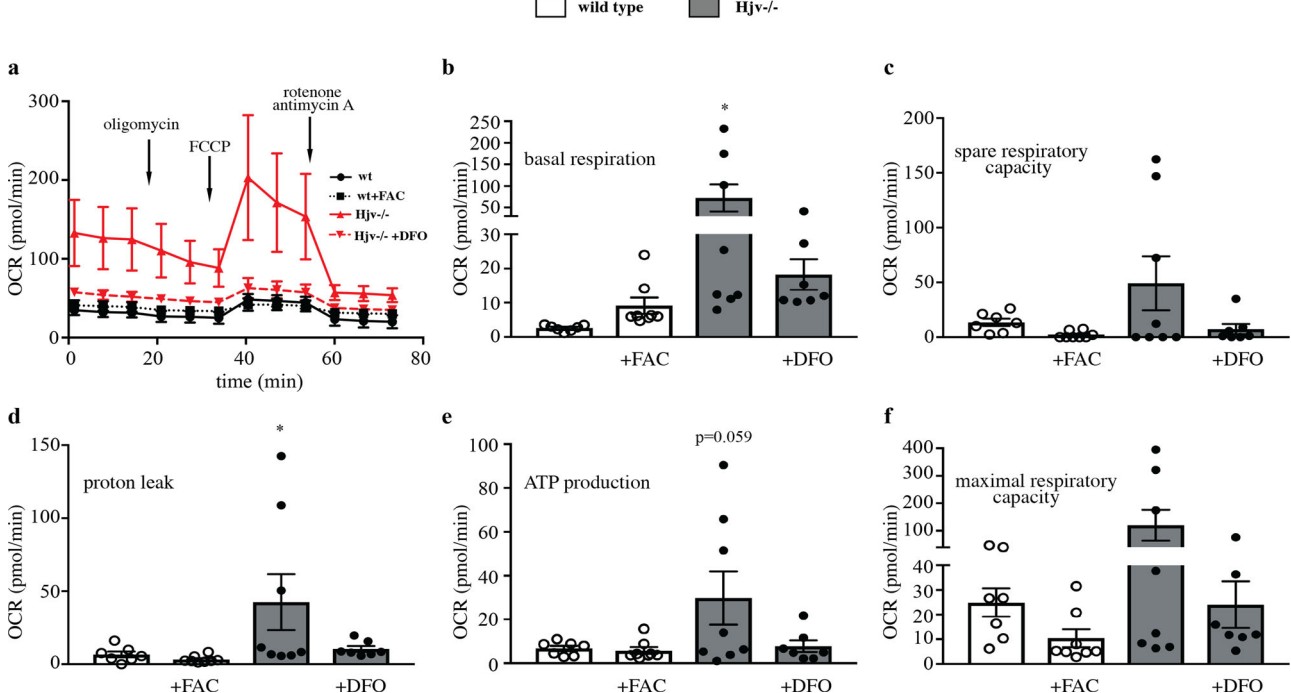

**Fig. 9 Hjv deficiency causes iron-dependent mitochondrial hyperactivity in hepatocytes.** Primary hepatocytes were isolated from wild type and Hjv$^{-/-}$ mice and left untreated or treated overnight with 15 μg/ml ferric ammonium citrate (FAC) or 100 μM desferrioxamine (DFO), respectively ($n = 7$–8 technical replicates for each condition). Subsequently, the cells were analyzed for oxygen consumption rate (OCR) using the Seahorse assay. **a** OCR following addition of oligomycin, FCCP and rotenone/antimycin A. **b** OCR for basal respiration. **c** OCR for spare respiratory capacity. **d** OCR for proton leak. **e** OCR for ATP production. **f** OCR for maximal respiratory capacity. Data are presented as the mean ± SEM. Statistical analysis was performed by one-way ANOVA. Statistically significant differences are indicated by *($p < 0.05$).

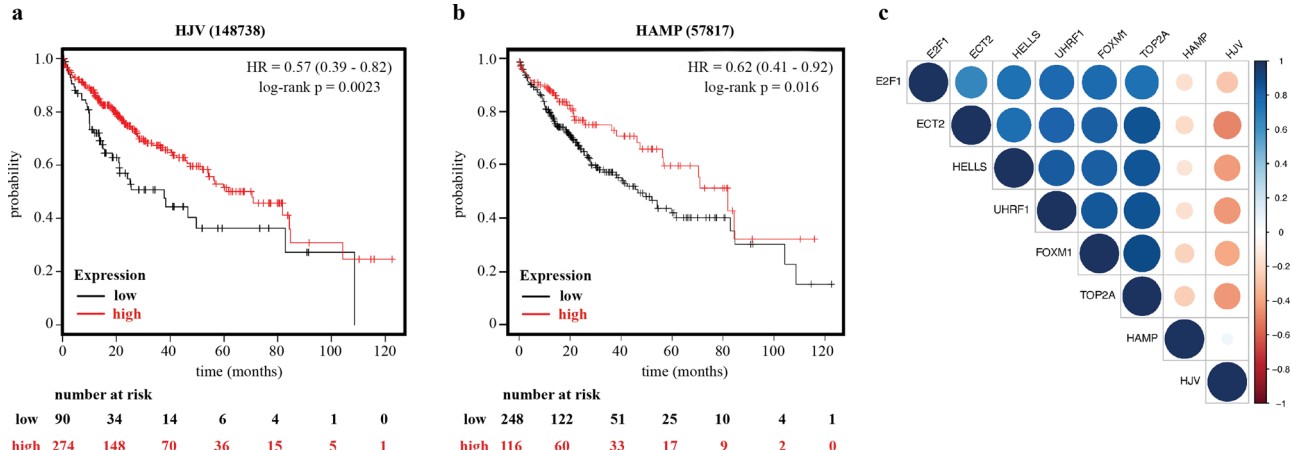

**Fig. 10 Expression of *HJV* and *HAMP* as a prognostic biomarker in HCC.** Survival analysis in 364 HCC patients plotted against expression of: **a** *HJV*; and **b** *HAMP*. **c** Spearman's rank correlation coefficients of expression levels among *HJV*, *HAMP* and the oncofetal markers *E2F1*, *ECT2*, *HELLS*, *UHRF1*, *FOXM1* and *TOP2A*. The size and more intense colors indicate stronger correlations (blue for positive and red for negative correlation). The plotting graph was designed using the corrplot package (https://github.com/taiyun/corrplot). All RNAseq data were obtained from the TCGA HCC dataset.

expression of Hamp or Hjv was calculated by the comparative Ct method. Data were normalized to Rpl19 and reported as fold increases compared to values from control mice.

**Seahorse assay.** Primary hepatocytes were isolated from wild type and Hjv$^{-/-}$ mouse livers, as previously described[55]. The hepatocytes were seeded to a Seahorse XF96 Cell Culture Microplate at a density of $2 \times 10^4$ cells/well and placed in a 5% $CO_2$ incubator at 37 °C. Some of the cells were left untreated and others were treated overnight with either 15 μg/ml ferric ammonium citrate (FAC) or 100 μM desferrioxamine (DFO). XFe96 sensor cartridge probes were immersed in the calibrant and stored in a 37 °C non-$CO_2$ incubator overnight. 8 mM glucose, 3 mM glutamine and 1 mM pyruvate were added to Agilent XF base medium, and the pH

was adjusted to 7.4 before cell media change. Each well was washed twice with the supplemented base media and the plate was incubated in a 37 °C non-$CO_2$ incubator for 1 h. Following addition of 1 μM oligomycin, 1.5 μM carbonyl cyanide-4-(trifluoromethoxy)phenylhydrazone (FCCP) and 1 μM rotenone/antimycin A to the designated drug loading ports, the cartridge and cell plate were loaded to a Seahorse XFe96 Analyzer to assess oxygen consumption rate (OCR). The complex V inhibitor oligomycin was used to derive ATP-linked respiration (by subtracting the oligomycin rate from baseline cellular OCR) and proton leak respiration (by subtracting non-mitochondrial respiration from the oligomycin rate). FCCP uncouples oxygen consumption from ATP synthesis and was used to induce maximal respiratory capacity. The complex III inhibitors rotenone and antimycin A were used to shut down the respiratory chain and reveal non-mitochondrial respiration.

**Statistical analysis of hematological and biochemical data**. Quantitative data were expressed as mean ± standard error of the mean (SEM). Multiple groups were subjected to analysis of variance (ANOVA) with Tukey's post-test comparison by using the GraphPad Prism software (v. 7.0e). A probability value $p < 0.05$ was considered statistically significant.

**Preparation of liver samples for proteomics**. Snap-frozen liver sections (including healthy tissue and cancerous lesions where applicable) were suspended in ice-cold extraction buffer (0.5 ml) and ground (using mortar and pestle) for 10 min. The extraction buffer consisted of 25 mM HEPES, (pH 8.0), 1.5 M urea, 0.02% Triton X-100, 5% (v/v) glycerol, 1 mM EDTA and 1:200 (v/v) protease inhibitor cocktail (Thermo Scientific, #78430). The suspension was centrifuged for 10 min at 15,000 g and the soluble protein supernatants were separated from the insoluble debris. Protein concentrations were measured by the DC-Protein Assay (BioRad). The resulting supernatant was used for reduction, alkylation and digestion. Samples were reduced by addition of 3 mM TCEP (tris(2-carboxyethyl) phosphine) for 45 min at room temperature then alkylated with 15 mM iodoacetamide for 60 min in the dark at room temperature. Proteolytic digestion was performed by addition of Trypsin/Lis-C solution, 500 ng (Promega, V5072) and incubated under shaking at 500 rpm at ambient temperature overnight. The digestion was stopped by addition of formic acid (1% final concentration) and centrifuged at 15,000 g for 2 min. The supernatant containing about 100 μg of digested protein was desalted on disposable TopTip C-18 columns (Glygen, #TT2C18.96) and dried by vacuum centrifugation.

**LC–MS/MS analysis**. All experiments were performed on an Orbitrap Fusion (Thermo Scientific) coupled to an Ultimate3000 nanoRLSC (Dionex). Peptides were separated on an in-house packed column (Polymicro Technology), 15 cm × 70 μm ID, Luna C18(2), 3 μm, 100 Å (Phenomenex) employing a water/acetonitrile/ 0.1% formic acid gradient. Samples were loaded onto the column for 200 min at a flow rate of 0.30 μl/min. Peptides were separated using 2% acetonitrile in the first 7 min and then using a linear gradient from 2–35% acetonitrile for 133 min, followed by a 35–98% acetonitrile gradient for 20 min, 98% acetonitrile for 15 min, 98–2% acetonitrile gradient for 10 min and a final 15 min wash with 2% acetonitrile. Eluted peptides were directly sprayed into mass spectrometer using positive electrospray ionization (ESI) at an ion source temperature of 250 °C and an ion spray voltage of 2.1 kV. The Orbitrap Fusion Tribrid was programmed in the data dependent acquisition mode. Full-scan MS spectra (m/z 350–2000) were acquired at a resolution of 60,000. Precursor ions were filtered according to monoisotopic precursor selection, charge state (+2 to + 7), and dynamic exclusion (70 s with a ± 10 ppm window). The automatic gain control settings were 5e5 for full FTMS scans and 1e4 for MS/MS scans. Fragmentation was performed with collision-induced dissociation (CID) in the linear ion trap. Precursors were isolated using a 2 m/z isolation window and fragmented with a normalized collision energy of 35%.

**MS data analysis**. MS raw files were analyzed using the MaxQuant software[56] in combination with the Andromeda search engine[57]. Peptides were searched against a mouse UniProt FASTA file containing 17,009 entries (30.12.2018) and a default contaminants database. Default parameters were used if not mentioned otherwise. N-terminal acetylation and methionine oxidation were set as variable modifications and cysteine carbamidomethylation was set as fixed modification. A minimum peptide length of 6 amino acids was required and False Discovery Rate (FDR) was set to 0.01 for both the protein and peptide level, determined by searching against a reverse sequence database. Enzyme specificity was set as C-terminal to arginine and lysine with a maximum of two missed cleavages. Peptides were identified with an initial precursor mass deviation of up to 10 ppm and a fragment mass deviation of 0.5 Da. The 'Match between runs' algorithm in MaxQuant[58] was enabled all samples to increase peptide identification rate. Proteins and peptides matching to the reverse database were discarded. For label-free protein quantitation (LFQ), a minimum ratio count of 2 was required[59].

**Downstream bioinformatics analysis**. All bioinformatic analyses including statistical analysis, database queries, and data visualization were performed in R using the pOmics R package available on GitHub. The proteinGroups files from MaxQuant were loaded into R and analyzed independently for the DEN and the Dietary dataset if not indicated otherwise. Proteins identified in the reverse database, as potential contaminants, or only by a modification site were removed. Normalized LFQ intensities were averaged between technical duplicates if both contained a value >0 and only proteins quantified in at least 25% (5 out of 20) of samples were considered for further analysis. Missing quantitative values were imputed from a normal distribution (down shift = 1.8, width = 0.3). Deregulated proteins between groups were identified by a one-way ANOVA and p-values were corrected by the Benjamini-Hochberg procedure. Individual differences between groups were filtered by Tukey's HSD test ($p < 0.05$) and a log2 fold-change difference greater than one standard deviation of all log2 fold-change values. Protein abundances were clustered using R's hclust function and visualized in heatmaps as Z-scored LFQ intensities using the ggplot2 and cowplot R packages. Functional enrichment was conducted using the clusterProfiler R package using annotations from the Gene Ontology and Reactome databases. Significantly enriched annotations of deregulated proteins and within clusters were identified by over-representation analysis and a Fisher's exact test (p-value < 0.05).

**Reporting summary**. Further information on research design is available in the Nature Research Reporting Summary linked to this article.

## Data availability
The source data underlying the graphs and charts in this study are provided in a Supplementary Data set. The raw proteomics data (MaxQuant protein groups) are available in the figshare repository with the identifier https://doi.org/10.6084/ m9.figshare.17185823.v1

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

## Acknowledgements

We thank Naciba Benlimame and Lilian Canetti for assistance with histology, and Dr. Cristina Baciu for generating the plot in Fig. 10c. We also thank Véronique Michaud for technical support at the Imaging and Phenotyping Core facility of the Lady Davis Institute. Dr. A. Allameh was granted a sabbatical leave from the Tarbiat Modares University, Tehran, Iran. E. Charlebois was supported by fellowships from the Natural Sciences and Engineering Research Council of Canada (NSERC) and the *Fonds de recherche du Québec – Santé* (FRQS). This work was funded by a grant from the Canadian Institutes of Health Research (CIHR; PJT-159730).

## Author contributions

A.A. conceptualization and experimental work; N.H. bioinformatics; E.C. experimental work; A.K. experimental work; W.G. experimental work; K.G. experimental work; E.P. bioinformatics; M.B. bioinformatics; Z.M. mass spectrometry; M.B. mass spectrometry; M.G. histopathology; C.F. experimental work, general management; K.P. conceptualization, supervision, manuscript writing

## Competing interests

The authors declare no competing interests.
