## [Transparent Peer Review File · Communications Biology]

Reviewers' comments:

Reviewer #1 (Remarks to the Author):

Analyses for the TCGA data was not presented.

Reviewer #2 (Remarks to the Author):

In this manuscript, the authors explore the role of the hemochromatosis protein hemojuvelin (HJV) and iron in the development of hepatocellular carcinoma (HCC). They provide data in mice demonstrating that HJV knockout mice are predisposed to develop HCC with aging or in response to the hepatic carcinogen DEN. They then undertake a series of mass spectrometry proteomics analyses to compare HJV knockout mouse livers with wildtype mice, without or with DEN treatment, or on different iron diets, as an unbiased approach to explore whether there are overlapping and/or independent effects of HJV and iron on the liver proteome that may be relevant to HCC development. Their studies suggest that HJV has both iron-dependent and iron-independent effects on the liver proteome. Finally, they demonstrate that reduced HJV and hepcidin expression are associated with poor prognosis in HCC patients. Overall, this is an intriguing study that provides new insights into a potential hepatoprotective effect of HJV against HCC, and serves to generate new hypotheses about potential mechanisms that may contribute. There are some questions that should be addressed by the authors to improve this study:

- 1) In the dietary manipulation studies, the authors report that control groups of mice were fed a standard diet. Were there any other differences between the standard diet and the iron-deficient or high-iron diets besides iron content? If they did not use matched purified diets in this study, how do the authors account for the other differences between these diets (besides iron) that may have influenced their proteomics data sets?
- 2) The HJV knockout mice on the iron-deficient diet still had significant iron overload. This may make it challenging to truly differentiate a role for iron overload and HJV loss in this study. Can the authors comment on this point?
- 3) To help corroborate whether HJV loss has both iron-dependent and iron-independent effects to predispose to HCC, does an iron-deficient diet impact the development of HCC in HJV knockout mice? What is the impact of a high-iron diet on HCC development in wildtype mice?
- 4) Could the finding that reduced HJV and hepcidin expression are associated with poor prognosis in HCC patients be merely a marker of less well-differentiated tumors? The findings in the mice do support a potential pathogenic role, but this should also be considered.

Reviewer #3 (Remarks to the Author):

The manuscript describes a study examining the effect of knocking out the gene encoding HJV on the development of HCC. HJV is essential for the correct expression of the iron regulatory hormone hepcidin and a deficiency in HJV results in reduced hepcidin expression and severe iron loading. In humans, this often leads to the development of HCC which is thought to be due to liver iron loading. In the current study, the authors have performed a series of proteomic analyses to examine the role of HJV disruption and/or iron loading in the development of HCC in mice. They find that the development of HCC is indeed linked to iron build up in the liver, however, suggest that HJV may also play an iron-independent role in HCC. If this can be validated, it would suggest novel pharmaceutical targets for treating patients with HCC.

Overall, the study appears to have been carefully carried out and conclusions are mostly supported by

the data. The manuscript is very well written and easy to follow, and there are detailed descriptions of all experimental procedures. However, I have several issues with the interpretation of the data as outlined below.

1. While the authors discuss the possibility of the distribution of iron explaining the "iron-independent" effect (paragraph beginning line 431), this needs to be acknowledged more throughout the manuscript. There is a distinct possibility that the different distribution between the models explains all of the differences seen when comparing loaded WT mice with deficient HJV KO mice. As the authors acknowledge in the Discussion, it is well known that the WT and WT loading models will favor Kupffer cell iron loading, while the HJV and HJV iron deficient models will favor hepatocyte loading with iron deficient Kupffer cells, particularly in HJV mice on the deficient diet. Because of this, I would like to see several statements made in the manuscript softened to acknowledge that, while iron-independent effects may be present, the studies presented do not conclusively show this. Examples are the Title, Abstract (lines 43-44), lines 83, lines 338-339 and lines 424-425.
2. Line 43...was this sentence referring to untreated HJV^{-/-} mice? If so, please state this.
3. Line 215...for the HJV^{-/-} mice and the mice treated with DEN, were the tissue samples analyzed made up of healthy liver tissue only or did they contain dysplastic foci and HCC? If it was the latter, does this affect the interpretation of the results i.e. does the data reflect changes occurring as a consequence of dysplastic foci being present rather than changes in healthy tissue that lead to foci development?
4. In the section entitled "HJV and HAMP expression in human HCC" (line 359) the authors state that lower expression of HJV and HAMP correlate with poor survival in HCC patients and suggest that targeting these proteins might be useful for improving outcome. However, this result could also reflect the fact that more aggressive cancers are less differentiated and, therefore, less likely to express the same proteins as healthy tissue. This would mean that low HJV and HAMP expression would be a consequence of a more aggressive tumor rather than contributing to it. This should be discussed.

We thank all reviewers for their constructive criticism. We have modified the title of the manuscript according to their comments. In addition, we shortened the abstract to 150 words, to comply with the journal's guidelines. Since the journal allows up to 10 illustrations, we moved one figure that contains significant information from supplemental to main (new Fig. 4). All modifications in the text of the revised manuscript are highlighted in red. A point-by-point response to the specific issues raised by the reviewers is provided below:

Reviewer 1

Q1: "The study is descriptive" and "more characterization of the mouse models, validation of the proteins identified and controls would benefit to supporting the authors conclusions".

We agree that the lack of validation of key findings was a major weakness of the previous submission. Our proteomics analysis identified several proteins and pathways that are linked to hepatocarcinogenesis, and the most prominent were mitochondrial. Instead of validating expression of single protein hits by Western or immunohistochemistry, we opted to focus on functional characterization of mitochondria. The data in new Fig. 8 demonstrate that HJV deficiency causes iron-dependent mitochondrial dysfunction in hepatocytes, strengthening our conclusions.

Q2: "Clinical data presented are of mRNA – but don't know how they chose the groups high or low?"

In response to this question, following sentence was added in the Results section: "To this end, we computed all possible cut off values between the lower and upper quartiles and selected the best performing threshold as cut off. In addition to the p-value, the False Discovery Rate (FDR) was computed to correct for multiple hypothesis testing (22)."

Q3: "Figures 1 and 4 can use higher resolution images".

We apologize for the low resolution. We followed the journals instructions for submission. Higher resolution images will be provided in the final version of the manuscript.

Reviewer 2

Q1: "Were there any other differences between the standard diet and the iron-deficient or high-iron diets besides iron content?"

There were no other differences in diets besides iron content. This is now clearly indicated in the Materials and Methods section.

Q2: "The HJV knockout mice on the iron-deficient diet still had significant iron overload. This may make it challenging to truly differentiate a role for iron overload and HJV loss in this study. Can the authors comment on this point?"

This important point is now addressed throughout the manuscript. Following specific suggestions of reviewer 3, who also raised similar concerns, we changed the title of the manuscript and modified several relevant statements in the Abstract, the Results section and the Discussion.

Q3: “To help corroborate whether HJV loss has both iron-dependent and iron-independent effects to pre-dispose to HCC, does an iron-deficient diet impact the development of HCC in HJV knockout mice? What is the impact of a high-iron diet on HCC development in wild type mice?”

We fully agree with this comment and propose these experiments as follow-up studies in the Discussion. We hope that the reviewer will concur that addressing this issue experimentally is beyond the scope of a revision.

Q4: “Could the finding that reduced HJV and hepcidin expression are associated with poor prognosis in HCC patients be merely a marker of less well-differentiated tumors?”

To address this important point, which was also raised by reviewer 3, we analyzed the HCC TCGA database for expression of oncofetal marker genes. The new data presented in Figs. 9C, S6 and Table S2 identify a weak but significant inverse correlation of HJV and HAMP with the oncofetal markers. We interpret these findings in the Discussion as partially supportive to the argument that low HJV and HAMP expression is a consequence of a more aggressive less differentiated tumor rather than a contributor.

Reviewer 3

Q1: “I would like to see several statements made in the manuscript softened to acknowledge that, while iron-independent effects may be present, the studies presented do not conclusively show this. Examples are the Title, Abstract (lines 43-44), lines 83, lines 338-339 and lines 424-425”.

Please, see response to Q2 of reviewer 2.

Q2: “Line 43...was this sentence referring to untreated HJV^{-/-} mice? If so, please state this.”

This is now stated in the modified sentence.

Q3: “Line 215...for the HJV^{-/-} mice and the mice treated with DEN, were the tissue samples analyzed made up of healthy liver tissue only or did they contain dysplastic foci and HCC? If it was the latter, does this affect the interpretation of the results i.e. does the data reflect changes occurring as a consequence of dysplastic foci being present rather than changes in healthy tissue that lead to foci development?”

We now indicate in the Materials and Methods that the liver sections used for proteomics analysis included both healthy tissue and cancerous lesions where applicable.

Q4: “...the authors state that lower expression of HJV and HAMP correlate with poor survival in HCC patients... However, this result could also reflect the fact that more aggressive cancers are less differentiated and, therefore, less likely to express the same proteins as healthy tissue.”

Please, see response to Q4 of reviewer 2.

REVIEWERS' COMMENTS:

Reviewer #1 (Remarks to the Author):

No further comments

Reviewer #2 (Remarks to the Author):

The authors have adequately addressed the concerns raised in the initial review.

Reviewer #3 (Remarks to the Author):

The authors have addressed my concerns.